# Young Nonalcoholic Wernicke Encephalopathy Patient Achieves Remission Following Prolonged Thiamine Treatment and Cognitive Rehabilitation

**DOI:** 10.3390/jcm12082901

**Published:** 2023-04-17

**Authors:** Erik Oudman, Jan W. Wijnia, Janice Bidesie, Zyneb Al-Hassaan, Sascha Laenen, Amy V. Jong-Tjien-Fa

**Affiliations:** 1Experimental Psychology, Helmholtz Institute, Utrecht University, 3584 CS Utrecht, The Netherlands; 2Slingedael Korsakoff Expertise Center, Lelie Care Group, 3086 EZ Rotterdam, The Netherlands; 3Daan Theeuwes Center for Intensive Neurorehabilitation, 3447 GN Woerden, The Netherlands

**Keywords:** Wernicke encephalopathy, Korsakoff’s syndrome, vitamin B1, nutrition, obesity surgery, cognitive rehabilitation, neuropsychological rehabilitation

## Abstract

Wernicke encephalopathy (WE), a neurological emergency commonly associated with alcohol use disorder, results from a severe deficiency of vitamin B1. If left untreated, patients either succumb to the illness or develop chronic Korsakoff’s syndrome (KS). Recently, an increasing number of nonalcoholic WE case studies have been published, highlighting a lack of understanding of malnutrition-related disorders among high-functioning patients. We present the case of a 26 year old female who developed life-threatening WE after COVID-19-complicated obesity surgery. She experienced the full triad of WE symptoms, including eye-movement disorders, delirium, and ataxia, and suffered for over 70 days before receiving her initial WE diagnosis. Late treatment resulted in progression of WE symptoms. Despite the severity, the patient achieved remission of some of the symptoms in the post-acute phase due to prolonged parenteral thiamine injections and intensive specialized rehabilitation designed for young traumatic brain injury (TBI) patients. The rehabilitation resulted in gradual remission of amnesia symptomatology, mainly increasing her autonomy. The late recognition of this case highlights the importance of early diagnosis and prompt, targeted intervention in the management of nonalcoholic WE, as well as underscores the potential for positive outcomes after delayed treatment through intensive cognitive rehabilitation in specialized treatment centers.

## 1. Introduction

Wernicke encephalopathy (WE) is a life-threatening neurological emergency caused by thiamine (vitamin B1) deficiency. Alcohol abuse, which often coincides with poor nutrition, leads to thiamine deficiency. Other thiamine deficiency-related conditions that can lead to WE are for example hyperemesis gravidarum, end-stage malignancies, inflammatory bowel disease, and psychiatric illness [1,2,3]. A relatively common cause of nonalcoholic WE is complicated bariatric surgery for obesity [4]. While the number of bariatric surgeries has been rising in the last decades, a lack of preassessment and aftercare, as well as noncompliance of patients that underwent obesity surgery, is relatively common [3,4,5]. Stress and uncertainty are frequently present in patients that undergo obesity surgery, and the restricted lifestyle that is required following surgery can increase these symptoms, leading to complex emotions related to food and eating [5]. Moreover, the restricted diet that is required after obesity surgery puts patients at risk for malnutrition [1,2,3,4,5].

Malnourished patients who have difficulty walking may display the presence of delirium, which can help in identifying WE [1]. Thiamine deficiency, the critical factor in the development of WE, can present with loss of appetite, dizziness, tachycardia, and urinary bladder retention, which are linked to anticholinergic autonomic dysfunction, as well as confusion or delirium [1,6,7], forming part of the classic triad of Wernicke encephalopathy [8]. The triad signs of Wernicke encephalopathy include ocular motility abnormalities such as external ophthalmoplegia and/or nystagmus, ataxia that mainly affects gait, and confusion or delirium are commonly used for its clinical diagnosis [1,9]. Caine et al. proposed four criteria for the clinical identification of WE: (i) the presence of dietary deficiencies, (ii) oculomotor abnormalities, (iii) cerebellar dysfunction, and (iv) either an altered mental state or mild memory impairment [10]. In addition to the risk of WE, thiamine deficiency is known to increase the risk of infections, such as pneumonia, urinary tract infections, abscesses, empyema, and sepsis with or without a known source [6]. Infections were reported in 35 out of 68 (51%) patients during the initial Wernicke phase [11], suggesting a complex relationship between malnourishment and infections.

Patients that have undergone obesity surgery are at a lifetime risk for malnutrition of micronutrients such as thiamine as a consequence of the procedure [3,12]. Importantly, patients who do not adhere to the recommended diets following obesity surgery are at even greater risk of developing diseases due to malnutrition. Lifelong postoperative clinical and laboratory monitoring are necessary to diagnose and treat malnourishment in postoperative obesity surgery patients [12]. A common medical complication of obesity surgery is a surgical site infection. Postoperative surgical-side infections are very common after obesity surgery (4.4-fold increased risk), and antibiotic prophylaxis is generally ineffective [13,14]. A possible underlying mechanism leading to this increased risk of infections is postoperative thiamine deficiency [11]. 

Moreover, a common complication of thiamine deficiency, as well as of obesity surgery, is vomiting. In almost all nonalcoholic cases that developed WE following any procedure or illness, vomiting and weight loss were the crucial factors highlighting malnourishment and prodromal WE [3,4]. Thiamine deficiency can be both a consequence and a cause of severe and prolonged vomiting [1,2]. Although some studies pointed out that thiamine deficiency is a relatively uncommon cause of WE [15], the cascade of vomiting leading to thiamine deficiency, leading to more vomiting and ultimately WE, is described in many cases that developed WE as a consequence of malnourishment [3,4].

While diagnosis of WE has been described as a complicated action, undertreatment of WE is also very common [16]. If WE is suspected, it is critical to immediately start treatment with intravenous thiamine, as orally administered thiamine is not adequate for preventing permanent brain damage. Timely administration of adequately dosed parenteral thiamine is a safe, inexpensive, and “brain-saving” treatment for Wernicke’s encephalopathy [17,18]. In the acute phase, treating the patient three times a day with 500 mg of thiamine intravenously until the symptoms resolve leads to the best outcome [8]. In case of severe malnourishment, supplementing magnesium is also necessary to increase thiamine uptake [1]. Unfortunately, insufficient thiamine treatment is common, such as oral thiamine supplementation or intravenous thiamine supplementation below 500 mg per day, leading to adverse outcome such as chronic immobility and ataxia (dependence on a wheelchair or walking aid), as well as more severe amnesia symptoms [1,2,3]. Recently, both acute and post-acute measures to increase the likelihood of positive outcome have been discussed, showing some positive outcomes of thiamine treatment, intensive physical therapy, structured day schedules, and involvement of a full team of specialists including occupational therapists, physicians, physical therapists, dietarians, and psychologists [19]. Extensive neuropsychological assessment is required at least 6 weeks following treatment of WE, and active efforts to ensure neuropsychological rehabilitation are an integral aspect of treating WE patients [19,20]. 

In the present case study, we describe a young female patient who underwent gastric sleeve surgery. Following the surgery, she developed COVID-19 during the acute phase, which complicated recovery. She showed signs and symptoms of WE following this infection, but the treatment delay was very long. After a prolonged stay in the hospital, she received intensive neurorehabilitation in a specialized center for young patients with traumatic brain injury. Ultimately, she was able to live independently without formal care 1.5 years later. 

## 2. Case Description

A 26 year old social worker in mental health care applied for a gastric sleeve in the obesity surgery clinic. After intake and screening, she underwent the gastric sleeve operation in October 2021. In the month following the procedure, she developed COVID-19 and was ill for over 1 week. After the COVID-19 infection, she developed nausea and vomiting. Her intake was minimal (<750 kcal) per day, caused by severe nausea and vomiting. She was bedridden in the weeks following initial infection. After developing loss of fine motor skills in her legs, she contacted her general practitioner. She was asked to take it easy for the following weeks. In the 10 weeks following her initial symptoms, she developed additional symptomatology, such as changed sensory perception in her arms and legs (feelings of pain and muscle contractions; also called polyneuropathy), loss of balance (truncal ataxia)m and loss of fine motor skills (ataxia in the arms and legs). Ultimately, she complained of double vision, issues with seeing light and dark, and confusion in the 2 weeks prior to admission to the hospital. After the entire trajectory of hospitalization and rehabilitation was completed, she explained that she contacted the physician and emergency care department over 30 times during the 10 weeks prior to actual admission to the hospital. Her impression was that her symptoms were considered to be caused by mental health issues, leading to improper treatment and care. Moreover, she lost 60 kg of weight during this time. As the cause of her symptoms was unclear, she felt increasingly anxious and expressed her hopelessness to the clinical professionals.

### 2.1. Hospitalization

In the beginning of March 2022, she presented at the hospital with the full triad of WE symptoms (delirium, nystagmus, and truncal ataxia or ataxia of the legs) and additional symptoms of late WE (bladder retention, loss of vision, extreme fatigue, muscle cramps, polyneuropathy including pain sensations in the arms and legs, and altered consciousness). She was diagnosed with WE by the division of internal medicine in the hospital. She was treated with 3 × 500 mg thiamine IV for 2 months. She received 0.5 mg of Haldol for the night, as she complained of vivid night terrors. She was fully dependent on care for her activities of daily living (ADLs), such as taking a shower, getting dressed, and going to the toilet, indicative of poor outcome [21]. During her stay in the hospital, an MRI was ultimately performed (30 March 2022), displaying no abnormalities (see Figure 1).

### 2.2. Consultation Korsakoff Expertise Center Slingedael (31 March 2022)

Korsakoff Expertise Center Slingedael was consulted to advise, facilitate aftercare, and admit the patient to the diagnostic facility of this center. During consultation, cognitive disorders were reported in the patient, such as attentional deficit and general confusion. At the time of consultation, she was unable to walk or stand without full support, indicative of severe ataxia. Throughout the consultation, she asked to go to the toilet at least 40 times, suggestive of memory issues, since she went to the toilet 5 min prior to the consultation. A urinary tract infection was suspected on the basis of reported pain, and later confirmed by the hospital. At the moment of consultation, she was fully dependent for ADL care. A diagnosis of active WE was suspected, and prolonged thiamine treatment was advised in accordance with earlier case reports showing optimal outcome after prolonged treatment [22]. Moreover, rehabilitation in a cognitive rehabilitation center was advised as aftercare according to the age and prognosis of WE in young patients [2,3]. After consultation, she was placed in a psychiatric ward until she could be admitted to the rehabilitation center.

### 2.3. Treatment in Daan Theeuwes Center for Intensive Rehabilitation

The patient was admitted to the Daan Theeuwes Center for Intensive Rehabilitation on 29 September 2022. The Daan Theeuwes Center for Intensive Neurorehabilitation, located in Woerden, the Netherlands, specializes in neurorehabilitation for adolescents and young adults (aged 16 to 35) who have experienced severe acquired brain injuries (ABI). The center offers intensive rehabilitation programs for both inpatients and outpatients. To be admitted, patients must be medically stable and conscious enough to participate in the program. The interdisciplinary treatment program is particularly intensive, consisting of 20–25 h of therapy per week, and is delivered by a team of experts including a physical medicine and rehabilitation physician, case manager, neuropsychologist, counselor, physical therapist, occupational therapist, speech therapist, and social worker.

In the Daan Theeuwes Center, the patient received intensive neurorehabilitation focused on regaining full ability to walk independently; she also received compensatory memory strategy training based on the training “remember it, do not forget it” consisting of at least 5 × 45 min per week of working with a notebook and schedule, learning how to make accurate notes of conversations to aid later recall, using mnemonics, and applying technological devices to support orientation in time and place. Moreover, she received compensatory training at least 3 × 45 min per week on learning to organize and plan information with tools such as planning strategy training, effective shifting skills, and actively managing complex tasks. Throughout rehabilitation, she made use of a customized Daan Theeuwes Center application on her mobile phone to keep a schedule and guide her through the building. Moreover, she could also review her treatment program, see her current goals, and see her progress regarding cognitive functioning. In February 2023, she was discharged with the intention of following the outpatient program for at least 6 weeks. 

#### Materials and Procedure

Neuropsychological examination was part of a routine assessment in the clinic, consisting of 4.5 days to test for cognitive impairments. Multiple measures of cognitive functioning were applied (see Table 1). The included neuropsychological tests were the Montreal Cognitive Assessment (MoCA) for general cognitive functioning [23], the Test of Memory Malingering for symptom validity assessment [24], the Balloons test for visual inattention [25], the Visual Object and Space Perception Test (VOSP) for perception [26], the DKEFS TMT for visual attention and flexibility [27], the DKEFS CWIT for attention and inhibition [28], the D2 Test for attention and concentration [29], the Rey Auditory Verbal Learning Test for verbal declarative memory (RAVLT) [30], the Rivermead Behavioral Memory Test Stories for verbal declarative memory, the Rey Complex Figures Test for visual memory [31], the Location Learning Test for visual declarative memory [32], Semantic and Phonemic Fluency [33], Subtests of the Behavioral Assessment Dysexecutive Syndrome (BADS) for executive functioning [34], and the Wechsler Adult Intelligence Scale-IV-NL (WAIS) for general intelligence [35]. She also followed the Measurement Feedback System for Intensive Neurorehabilitation (MFSIN) [36], including the 6 min Walk Test for physical condition [37], the Berg Balance Scale for physical balance [38], the 10 m Walking Test for physical condition [39] and the supervision rating scale for functional autonomy [40]. Some results are displayed in Figure 2 and Figure 3. The results in Figure 2 and Figure 3 show an increased conditional state and better balance scores, indicative of more autonomous functioning.

### 2.4. Second Consultation

On 21 February 2023 the second consultation was performed. The patient was released from the clinic 1 week prior and was able to live independently with minimal support. She also managed to get her driver’s license back after completing assessment. Within this follow-up consultation, she stated that she still had some memory complaints, but they were drastically reduced compared to the onset of WE. For example, at the start, she could not remember what she did 5 min prior in the hospital, whereas, at the start of her rehabilitation, she only managed to remember information from the previous hour. She is now able to give a review of the full day. She still misses some details, but uses a notebook to write down important information. She also managed to keep in touch with some of the patients she met when she was an inpatient. She indicated that preventing WE in obesity surgery patients is of the utmost importance. On 15 February 2023, 6 days prior to the second consultation, she scored 28 out of 30 points on the MoCA, indicative of cognitive improvement compared to 7 November 2022 (see Table 1, score 22/30; not statistically significant compared to normative data) [41]. 

## 3. Discussion

The patient in the present case report developed WE following complicated obesity surgery. Prodromal, acute, and late symptomatology of WE was missed, leading to a long treatment delay with end-stage characteristics of WE. While such a severe course of WE can lead to severe chronic Korsakoff’s syndrome or even chronic WE [42,43], this patient showed remission of acute symptoms through extensive physical and neuropsychological rehabilitation and prolonged thiamine treatment. The present study highlights the importance of regular vitamin checks after obesity surgery. Moreover, it highlights the importance of intensive rehabilitation in the post-acute phase, specifically for young patients with WE, to optimize functioning.

WE is a life-threatening condition following acute thiamine deficiency. Many studies have focused on acute treatment with high doses of parenteral thiamine in WE, but treatment delay is common [1,2,3,16]. Importantly, late treatment often leads to more severe outcome suggesting that additional focus is needed on the prodromal and early phase of WE [1,2,3]. In the present case study, the patient contacted the general practitioner 30 times and she lost 60 kg prior to admission to the hospital. Her vitamin B1 status was missed, and, due to COVID-19 infection and legislations, she had not received active follow-up after obesity surgery. As earlier evidence suggests, patients have a lifelong risk for WE following obesity surgery, and follow-ups are generally missed [4]. Guidelines for quality control in obesity surgery require adequate vitamin supplementation and follow-up after surgery, as well as specific focus on patients that show complications leading to vomiting.

Parenteral thiamine replacement treatment is necessary to treat WE. As concluded in a recent systematic review, the optimal dosing regimen for WE is not established [44]. Often, doses of 100 mg/day are described in the acute phase, but higher doses are likely to be more beneficial as indicated by retrospective chart reviews [19]. Specifically, cognitive and motoric outcomes are better in groups receiving more than 500 mg/day as an initial dose. Thomson et al. [45] pointed to a dosing regime of 500 mg three times per day until symptoms resolve showing the most favorable outcome. High doses of thiamine injections are well tolerated and often do not lead to side-effects. In the present case, no side-effects were observed, and high doses resulted in ultimate remission of WE symptoms. 

Prolonged parenteral thiamine treatment following WE in nonalcoholic cases is relatively uncommon practice, but seems to have additional benefits regarding cognitive and motoric outcome [22]. In the present case study, the patient did receive 2 months of parenteral thiamine suppletion, supporting her recovery. It would be relevant to investigate longer treatment regimens prospectively in new studies on WE outcome to validate the potential of longer parenteral thiamine treatment in more WE patients.

Infections are common in WE [11]. The present case developed COVID-19, increasing the risk of thiamine depletion and WE. Recently, a number of papers focused on the increased likelihood of developing WE following COVID-19 infections [46,47,48]. A shortage in thiamine is a relatively common complication of COVID-19 infections. In the present case, the COVID-19 infection was the main complication leading to prolonged vomiting and thiamine depletion, ultimately leading to WE. In the hospital, viral loads of the patient were determined as normal, suggesting that the symptoms were not directly caused by long COVID-19 [49].

As earlier research suggests, medical services are not fully equipped to treat young patients with WE of nonalcoholic origin [17]. The current case study illustrates that intensive neurorehabilitation is required for nonalcoholic WE patients, especially when they are young. Brain plasticity might be better preserved in young WE patients. Moreover, executive deficits tend to be smaller in this group. Earlier studies on memory rehabilitation specifically pointed out to the beneficial effects of errorless learning [50] and assistive technology [51] for patients diagnosed with chronic KS. In the present case study, both compensatory techniques and assistive technology were applied in the rehabilitation of the patient.

In conclusion, the present case study illustrates that complicated obesity surgery can lead to WE. Vomiting and infections caused a rapid loss of weight and eventually a full WE triad. Preventive thiamine treatment as a part of aftercare was not given, leading to further deterioration. Intensive neurorehabilitation resulted in recovery of function, following prolonged thiamine supplementation.

## Figures and Tables

**Figure 1 jcm-12-02901-f001:**
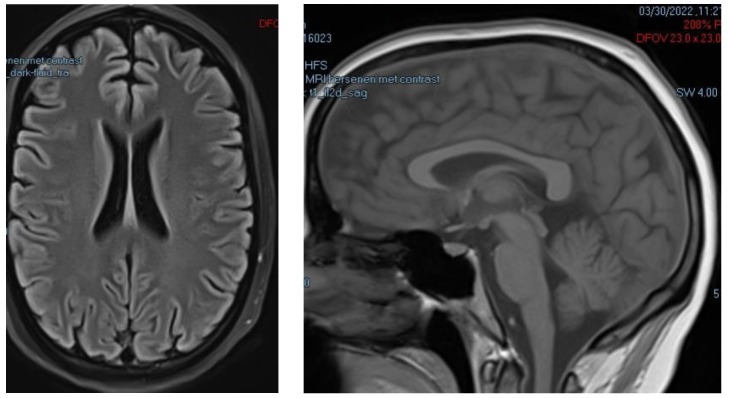
MRI scan of the brain during hospitalization 30 March 2022: axial view (**left**), sagittal view (**right**). No alterations are visible.

**Figure 2 jcm-12-02901-f002:**
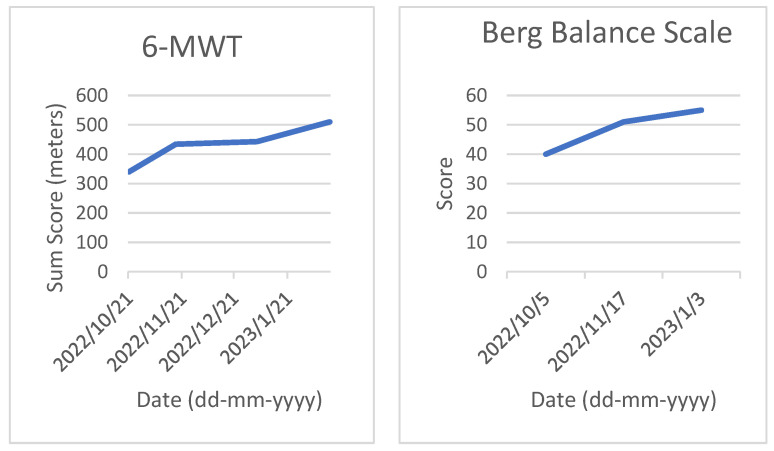
Task performance on the 6 min Walking Test (6-MWT, **left**) and Berg Balance Scale over time (**right**).

**Figure 3 jcm-12-02901-f003:**
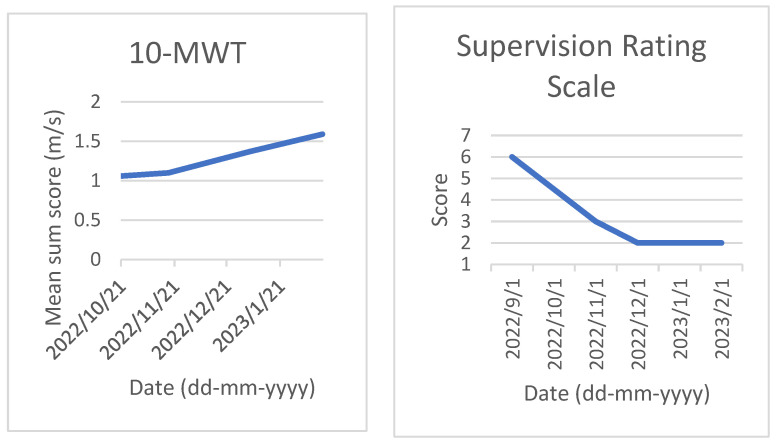
Task performance on the 10 m Walking Test (10-MWT, **left**) and Supervision Rating Scale (**right**).

**Table 1 jcm-12-02901-t001:** Neuropsychological examination of the patient on 7 November 2022.

Test	Measurement	Score	Norms	Analysis
MoCA (7.3)	General cognition	22/30	≤26	Impaired
Test of Memory Malingering	Symptom validity testing	Trial 2: 50	≥45	Normal
Balloons Test	Visual inattention	Score A: 10 + 10 = 20	≥17	Normal
		Score B: 10 + 9 = 19	≥45%	
VOSP	Object and space perception		Normal	Normal
DKEFS TMT	Attention	1: 16 s, 0 errors	W = 12	Average
	“	2: 26 s, 0 errors	W = 11	
	“	3: 23 s, 0 errors	W = 12	
	Mental flexibility	4: 48 s, 0 errors	W = 12	
	“	5: 26 s, 0 errors	W = 10	
DKEFS CWIT	Attention	1: 25 s, 0 errors	W = 11	Average
	“	2: 21 s, 0 errors	W = 11	
	Inhibition	3: 51 s, 0 errors	W = 10	
	“	4: 60 s, 0 errors	W = 9	
D 2 Test	Concentration	Total: 436	T = 44	Average
	“	Total errors: 30	T = 41	Low average
	“	Error percentage: 6%	T = 42	Low average
	“	Concentration: 149	T = 43	Low average
	“	Variation: 11	T = 41	Low average
RAVLT	Verbal declarative memory	9-11-10-13-14	T = 50	Average
	“	Recall: 9	T = 35	Low
	“	Recognition 30/30		
RBMT stories	Verbal declarative memory	Total (A + B) = 8.5 + 9 = 17.5	T = 42	Low average
	“	Total (A + B) = 4 + 5.5 = 9.5	T = 30	Low
	“	Remembered 54%	T = 23	Very low
Rey Complex Figures	Visuoconstruction	Immediate (3 min.): 12/36	P < 1	Very low
	Visual memory	Delayed (30 min.): 12.5/36	P < 1	Very low
	Delayed visual memory	Recognition: 21/24	P = 38	Average
Location Learning Test	Visuospatial declarative memory	Total error (4-3-0-1-1) = 9	P = 29	Average
	“	Learning index = 0.31	P = 1	Very low
	“	Delayed remembering = 1	P = 93	High
Fluency				
Animals	Sematic fluency	25	T = 48	Average
Occupations	“	19	T = 48	Average
Supermarket	“	24	T = 48	Average
D	Executive fluency	12	T = 45	Average
A	“	13		
T	“	11		
BADS	Planning- and organizing skills	Zoo map test: 9/16	PS 2 à z = −0.39	Average
	Planning skills	Key search test: 12/16	PS 3 à z = 0.3	Average
	“	Action plan test: 5/5	PS 4 à z = 0.44	Average
WAIS-IV-NL	General intellectual functioning	Total IQ: 93	Average	Average

MoCA = Montreal Cognitive Assessment, VOSP = Visual Object and Space Perception, D-KEFS = Delis-Kaplan Executive Function System, TMT = Trail Making Test, CWIT = Color Word Interference Test, RAVLT = Rey Auditory Verbal Learning Test, RMBT = Rivermead Behavioral Memory Test, BADS = Behavioral Assessment Dysexecutive Syndrome, WAIS-IV-NL = Wechsler Adult Intelligence Scale Dutch Version.

## Data Availability

Data supporting results are saved on the server of the University of Utrecht. Due to patient confidentiality, raw data are not publicly available.

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
