# Peer review of "Young Nonalcoholic Wernicke Encephalopathy Patient Achieves Remission Following Prolonged Thiamine Treatment and Cognitive Rehabilitation"

_jcm, 2023, doi:10.3390/jcm12082901_

Round 1

Reviewer 1 Report

The paper titled “Young Non-Alcoholic Wernicke Encephalopathy Patient Achieves Remission Following Prolonged Thiamine Treatment and Cognitive Rehabilitation” written by Oudman and colleagues is a clinical case that analyzes the relationship between WE, caused by weight loss surgery and Covid-19 infection, and thiamine supplementation.

1. Correct the word “total” in table 1.

2. Is the label (sscore) of the axis in figure 3 correct?

3. Homogenize the date format in the manuscript. I suggest day/month/year format.

4. I suggest that the authors´ contribution be specific regarding the role of each author in the work. I also recommend that the role of each author be stated in this section and not in the text of the manuscript since the role of the second and third authors is missing, at least in the experimental protocol, but the investigative role of the fourth, fifth and sixth authors was not specified in the authors' contribution section.

5. Was Covid-19 viral load determined during the study? because it has been reported that a consequence of Covid-19 infection is also cognitive impairment that occurs long after infection (DOI:https://doi.org/10.1016/j.eclinm.2021.1010)

6. Please briefly describe the methods used to assess the reported cognitive function (e.g. MoCA)

7. Was the study approved by an ethics committee?

Author Response

Response to reviewer 1

  1. Correct the word “total” in table 1.

Response: We have corrected the word “total” in table 1

  1. Is the label (sscore) of the axis in figure 3 correct?

Response: We have corrected this issue.

  1. Homogenize the date format in the manuscript. I suggest day/month/year format.

Response: We have corrected the date format in according to the reviewers request.

  1. I suggest that the authors´ contribution be specific regarding the role of each author in the work. I also recommend that the role of each author be stated in this section and not in the text of the manuscript since the role of the second and third authors is missing, at least in the experimental protocol, but the investigative role of the fourth, fifth and sixth authors was not specified in the authors' contribution section.

Response: We thank the reviewer for this suggestion, removed all the author’s contribution in      the written text and included the contributions after the discussion section.

  1. Was Covid-19 viral load determined during the study? because it has been reported that a consequence of Covid-19 infection is also cognitive impairment that occurs long after infection (DOI:https://doi.org/10.1016/j.eclinm.2021.1010)

Response: We are aware that Covid-19 infections are able to induce more (persistent) neurocognitive problems than eventually thought. In the hospital viral load was determined as within normal parameters (part of routine investigation). We now include the referenced report in our manuscript in the discussion section.

  1. Please briefly describe the methods used to assess the reported cognitive function (e.g. MoCA):

Response: We thank the reviewer for this suggestion and included a description of testing methods in our methods section, and include the measurement domains in the table.

  1. Was the study approved by an ethics committee?

Response: We now include this as a statement in the revised manuscript.

Reviewer 2 Report

Dear authors,

I have read through the paper carefully and the results might be encouraging overall and of scientific interest. This paper may give an important contribution in highlighting the importance of an early diagnosis in non-alcoholic Wernicke encephalopathy and the prompting of targeted intervention in the management of the disease. However, there are some methodological issues which the authors should take into account as I discussed below.

As far as I am concerned, the paper could give an important contribution if the authors fulfill these shortcomings.

1)    The authors reported a case of a young women with Wernicke encephalopathy. Did the patient receive a diagnosis according to an international classification of the disease (es. ICD-9 or 10)? Who provided the diagnosis?

2)    Did the patients sign an informed consent?

3)    Is there an ethical approval?

4)    The patient underwent to an extensive neuropsychological assessment once she was admitted to the Center for intensive Rehabilitation. However, the authors did not report a further assessment after the cognitive treatment. The aim of a second neuropsychological assessment -after the cognitive rehabilitation-would have provided objective evidence of the patient’s improvement.

5)     The authors did not perform a statistical analysis in order to test the significance of remissions of symptoms.

Minor revisions

1)    In the introduction paragraph more evidence/studies about the positive effects of thiamine treatments and cognitive rehabilitation in this population should be provided.

2)    A “materials” section should be included in the test. In this section the authors could provide a better description of neuropsychological and motor tests and their score.

3)    Table 1 should be rearranged. For example, tests could be clustered for the cognitive domain/process they are intended to assess.

4)    Self-citations are the 21% of the total reference. I would suggest either to remove some self-citations or extend the number references.

Author Response

Dear authors,

I have read through the paper carefully and the results might be encouraging overall and of scientific interest. This paper may give an important contribution in highlighting the importance of an early diagnosis in non-alcoholic Wernicke encephalopathy and the prompting of targeted intervention in the management of the disease. However, there are some methodological issues which the authors should take into account as I discussed below.

Response: Thank you for this compliment.

  • The authors reported a case of a young women with Wernicke encephalopathy. Did the patient receive a diagnosis according to an international classification of the disease (es. ICD-9 or 10)? Who provided the diagnosis?

Response: The division of internal medicine confirming the diagnosis of Wernicke Encephalopathy in the hospital did not explicitly mention the use of ICD-9 or ICD-10 classification. She received the diagnosis by the internist in the hospital. The diagnosis is now mentioned in the revision.

2)    Did the patients sign an informed consent?

Response: See the informed consent statement at the bottom of the manuscript.

3)    Is there an ethical approval?

Response: See the ethical approval statement at the bottom of the manuscript.

4)    The patient underwent to an extensive neuropsychological assessment once she was admitted to the Center for intensive Rehabilitation. However, the authors did not report a further assessment after the cognitive treatment. The aim of a second neuropsychological assessment -after the cognitive rehabilitation-would have provided objective evidence of the patient’s improvement.

Response: in the revised manuscript we included a recent MoCA score to highlight improvement of cognitive functioning. She managed to score 28/30 points in comparison to 22/30 points at admission: On 15-2-2023, six days prior to the second consultation, she scored 28 out of 30 points on the MoCA, indicative of cognitive improvement compared to 7-11-2022 (see table 1, score 22/30); and not statistically significant compared to normative data.

5)     The authors did not perform a statistical analysis in order to test the significance of remissions of symptoms.

Response: While paired t-tests are often used as a statistical analysis for 2 means, in the present study we only have a couple of measurements per index. This complicates statistical testing for many of the applied scores. Regarding the MoCA score, the initial MoCA score was statistically different from an normative group of higher educated individuals (Borland et al., 2017; z-score = -4.54545) and the MoCA was statistically not significant in follow-up (Z-score = 0.90909). We report this in the revised manuscript: On 15-2-2023, six days prior to the second consultation, she scored 28 out of 30 points on the MoCA, indicative of cognitive improvement compared to 7-11-2022 (see table 1, score 22/30); and not statistically significant compared to normative data.

Minor revisions

  • In the introduction paragraph more evidence/studies about the positive effects of thiamine treatments and cognitive rehabilitation in this population should be provided.

Response: We now include more evidence in our revised introduction section regarding the outcomes of thiamine treatment: While diagnosis of WE has been described as a complicated action, also un-der-treatment of WE is very common [16]. If WE is suspected, it is critical to immedi-ately start treatment with intravenous thiamine, as orally administered thiamine is not adequate for preventing permanent brain damage. Timely administration of ade-quately dosed parenteral thiamine is a safe, inexpensive, and “brain-saving” treatment for Wernicke’s encephalopathy [17, 18]. In the acute phase, treating the patient three times a day with 500 mg of thiamine intravenously until the symptoms resolve, leads to the best outcome [8]. In case of severe malnourishment, also supplementing magne-sium is necessary to increase thiamine uptake [1]. Unfortunately insufficient thiamine treatment is common, such as oral thiamine supplementation or intravenous thiamine supplementation below 500 mg per day, leading to adverse outcome such as chronic immobility and ataxia (dependence on a wheelchair or walking aid), and more severe amnesia symptoms [1, 2, 3]. Recently, both acute and post-acute measures to increase the likelihood of positive outcome have been discussed, showing some positive out-comes of thiamine treatment, intensive physical therapy, structured day-schedules and involvement of a full team of specialists including occupational therapists, physicians, physical therapists, dietarians, and psychologists [19]. Extensive neuropsychological assessment is required after at least six weeks following treatment of WE, and active efforts to ensure neuropsychological rehabilitation are an integral aspect of treating WE patients [19, 20].

  • A “materials” section should be included in the test. In this section the authors could provide a better description of neuropsychological and motor tests and their score.

Response: We now include a section on the neuropsychological and motor tests in a materials and procedure section: “Materials and procedure

Neuropsychological examination was part of a routine assessment in the clinic, consisting of four half days to test for cognitive impairments. Multiple measures of cognitive functioning were applied (see table 1). The included neuropsychological tests were the Montreal Cognitive Assessment (MoCA) for general cognitive functioning [23],  the Test of Memory Malingering for symptom validity assessment [24], the Balloons test for visual inattention [25], the Visual Object and Space Perception Test (VOSP) for perception [26], the DKEFS TMT for visual attention and flexibility [27], the DKEFS CWIT for  attention and inhibition [28], the D2 Test for attention and concentration [29], the Rey Auditory Verbal Learning Test for verbal declarative memory (RAVLT) [30], the Rivermead Behavioral Memory Test Stories for verbal declarative memory, the Rey Complex Figures Test for visual memory [31], the Location Learning Test for visual declarative memory [32], Semantic and Phonemic Fluency [33], Subtests of the Behavioural Assessment Dysexecutive Syndrome (BADS) for executive functioning [34], and the Wechsler Adult Intelligence Scale-IV-NL (WAIS) for general intelligence [35]. She also followed the Measurement Feedback System for Intensive Neurorehabil-itation (MFSIN) [36], including the 6 Minute Walk Test for physical condition [37], the Berg Balance Scale for physical balance [38], the 10 Minute Walking Test for physical condition [39] and the supervision rating scale for functional autonomy[40]. Some re-sults are displayed in Figure 2 and 3. The results in Figure 2 and 3 show an increased conditional state and better balance scores indicative of more autonomous functioning”.

3)    Table 1 should be rearranged. For example, tests could be clustered for the cognitive domain/process they are intended to assess.

Response: We now include the cognitive domains of each test in the table.

4)    Self-citations are the 21% of the total reference. I would suggest either to remove some self-citations or extend the number references.

Response: We would like to apologize for the high number of self-refences in the original manuscript. As we published six systematic reviews on non-alcoholic WE in the past five years, it was a bit of an easy fix to include some self-references. In the revised manuscript we replaced some of the self-references with papers written by other authors, and we included more references in our introduction and discussion section.

Reviewer 3 Report

The authors present an interesting case study of a 26-year-old female who developed Wernicke's Encephalopathy (WE) following Covid-19 complicated obesity surgery. She presented with the typical triad of WE symptoms, including eye-movement disorders, delirium, and ataxia. She did not receive a WE diagnosis until over 70 days post surgery and this resulted in a delay of treatment and a progression of WE symptoms.  The authors report that the patient achieved remission of some of her symptoms in the post-acute phase as a result of prolonged parenteral thiamine injections and intensive specialized rehabilitation designed for young Traumatic Brain Injury (TBI) patients. There was one question that the authors should address. Specifically, the authors report that this patient was treated three times a day with 500mg of thiamine received intravenously. This is a higher dose than is often used. The authors do point out that the total 1500mg dose, in conjunction with neurorehabilitation, was effective for the patient. It may be interesting for the authors to write a bit as to why lower doses of thiamine are typically given and to mention whether their patient experienced any side effects from the higher dose. All in all, this was a very interesting case study which raised a very important point as regards under-dosing thiamine in cases of WE and the importance of including neurorehabilitation as part of the treatment regimen.

Author Response

The authors present an interesting case study of a 26-year-old female who developed Wernicke's Encephalopathy (WE) following Covid-19 complicated obesity surgery. She presented with the typical triad of WE symptoms, including eye-movement disorders, delirium, and ataxia. She did not receive a WE diagnosis until over 70 days post surgery and this resulted in a delay of treatment and a progression of WE symptoms.  The authors report that the patient achieved remission of some of her symptoms in the post-acute phase as a result of prolonged parenteral thiamine injections and intensive specialized rehabilitation designed for young Traumatic Brain Injury (TBI) patients.

Response: Thank you.

There was one question that the authors should address. Specifically, the authors report that this patient was treated three times a day with 500mg of thiamine received intravenously. This is a higher dose than is often used. The authors do point out that the total 1500mg dose, in conjunction with neurorehabilitation, was effective for the patient. It may be interesting for the authors to write a bit as to why lower doses of thiamine are typically given and to mention whether their patient experienced any side effects from the higher dose.

Response: Thank you again. We include an additional section (in the revised manuscript) in our discussion on the dosing and effects: “Parenteral thiamine replacement treatment is necessary to treat WE. As a recent systematic review concludes, the optimal dosing regimen for WE is not established [44]. Often doses of 100mg/day are described in the acute phase, but higher doses are likely to be more beneficial as indicated by retrospective chart reviews [19]. Specifical-ly cognitive and motoric outcome are better in groups receiving more than 500mg/day as an initial dose.  Thomson et al [45] pointed to a dosing regime of 500mg/3 times per day until symptoms resolve showing the most favorable outcome. High doses of thia-mine injections are well-tolerated and often do not lead to side effects. In the present case, no side effect were observed, and high doses resulted in ultimate remission of WE symptoms”.

All in all, this was a very interesting case study which raised a very important point as regards under-dosing thiamine in cases of WE and the importance of including neurorehabilitation as part of the treatment regimen.

Response: Thank you.

Round 2

Reviewer 2 Report

Dear Author,

I appreciated the responses to the previous reviews and I believe the manuscript may be taken into consideration for a potential publication. However, I have one minor concern about the MoCA's normative data. In reference 41 you referred to normative data coming from a Swedish population. As I may infer from the authors' affiliation and from the "Korsakoff Expertise Center Slingedael" the patient was Dutch. Accordingly, the authors should consider the normative data gathered from a Dutch sample (e.g. Bruijnen et al., 2020, Psychometric properties of the Montreal Cognitive Assessment (MoCA) in healthy participants aged 18–70).

Author Response

We agree with the reviewer, and now included the Bruijnen et al. 2020 reference instead of the Swedish reference as reference 41.